# Investigation of Structural Degradation of Fiber Cement Boards Due to Thermal Impact

**DOI:** 10.3390/ma12060944

**Published:** 2019-03-21

**Authors:** Zbigniew Ranachowski, Przemysław Ranachowski, Tomasz Dębowski, Tomasz Gorzelańczyk, Krzysztof Schabowicz

**Affiliations:** 1Experimental Mechanics Division, Institute of Fundamental Technological Research, Polish Academy of Sciences, Pawińskiego 5B, 02-106 Warszawa, Poland; pranach@ippt.pan.pl (P.R.); tdebow@ippt.pan.pl (T.D.); 2Faculty of Civil Engineering, Wrocław University of Science and Technology, Wybrzeże Wyspiańskiego 27, 50-370 Wrocław, Poland; Tomasz.Gorzelanczyk@pwr.edu.pl (T.G.); krzysztof.schabowicz@pwr.edu.pl (K.S.)

**Keywords:** cement-based composites, fiber cement boards, durability, ultrasound measurements

## Abstract

The aim of the present study was to investigate the degradation of the microstructure and mechanical properties of fiber cement board (FCB), which was exposed to environmental hazards, resulting in thermal impact on the microstructure of the board. The process of structural degradation was conducted under laboratory conditions by storing the FCB specimens in a dry, electric oven for 3 h at a temperature of 230 °C. Five sets of specimens, that differed in cement and fiber content, were tested. Due to the applied heating procedure, the process of carbonization and resulting embrittlement of the fibers was observed. The fiber reinforcement morphology and the mechanical properties of the investigated compositions were identified both before, and after, their carbonization. Visual light and scanning electron microscopy, X-ray micro tomography, flexural strength, and work of flexural test *W_f_* measurements were used. A dedicated instrumentation set was prepared to determine the ultrasound testing (UT) longitudinal wave velocity *c*_L_ in all tested sets of specimens. The UT wave velocity *c*_L_ loss was observed in all cases of thermal treatment; however, that loss varied from 2% to 20%, depending on the FCB composition. The results obtained suggest a possible application of the UT method for an on-site assessment of the degradation processes occurring in fiber cement boards.

## 1. Introduction

Fiber cement board (FCB) is a versatile, green, and widely-applied building material. It acts as a substitute for natural wood and wood-based products, such as plywood or oriented strand boards (OSB). The properties of FCB, as a construction material, make it preferable for use as a ventilated, façade cladding for newly-built and renovated buildings, interior wall coverings, balcony balustrade panels, base course and chimney cladding, and enclosure soft-fit lining [1]. FCB can be applied to unfinished, painted, or simply-impregnated surfaces. Fiber cement components have been used in construction for over 100 years, mainly as roofing covers, in the form of corrugated plates or non-pressurized tubes. FCB façades are exposed to a variety of different environmental hazards. Adverse factors can include visual an ultraviolet light radiation, wind and ice-clod impacts, and thermal stresses evoked by temperature changes, etc. [2,3]. These hazards may result in board embrittlement, shrinkage, or bending. An example of a FCB façade, showing considerable damage after ten years of exposure to climate hazards, is presented in Figure 1. The fracture process that can develop in building materials is complex because the strains are not uniformly distributed during the fracture, particularly in regions where there are cracks. The facade boards are usually fixed to the wall- construction on their edgings, which exposes them to flexural stresses. The currently-applied fiber cement boards are designed to carry the mechanical load by the cellulose and polyvinyl alcohol (PVA) fiber reinforcements. The fibers reinforce the FCB component only when they are added in a specific quantity (5–10% wt.) and when they are uniformly dispersed throughout the cementitious matrix. A highly complex procedure is required to achieve this goal, as well as to avoid faults under efficient industrial conditions. Hatschek solved the problem by inventing a machine with a rotating sieve and a vat containing a diluted fiber slurry, Portland cement, and mineral components [4,5]. A thin film of FCB is formed on a moving belt, partially wrapped around the sieve, similar to the procedure used in paper sheet-making [6].

As the service performance of fiber cement boards may be affected by the improper function of reinforcements (i.e., damaged fibers, inhomogeneous concentrations, or poor quality fibers), several methods were proposed for testing the performance of the boards. These include:-emitting and receiving the ultrasonic Lamb waves [7,8],-the impact-echo method combined with the impulse-response method [9],-the ultrasound (UT) longitudinal wave velocity *c*_L_ method [10,11],-the acoustic emission method [12],-X-ray micro tomography [13,14].

The UT wave velocity *c*_L_ method is one of the few methods that can be applied in situ to control the degradation processes in building façades. Some authors [15,16] have performed experiments that resulted in the degradation of fiber reinforcements by pyrolyzing the fibers; i.e., by exposing them to an elevated temperature for several hours. These authors reported that a loss of material elasticity can be observed when thermal treatment is conducted using temperatures exceeding 100 °C.

It is also possible to analyze the morphology and distribution of fibers in the cementitious matrix using microscopy, by applying different kinds of visible light during the testing process. A detailed description of this procedure can be found in [17]. Cellulose fibers are usually thicker than PVA fibers. They are beige-colored, like wood, while PVA fibers are pale and transparent. After thermal treatment, both types of fibers become brown, which is evidence of their structural dehydrogenation, so that what remains in the fibers is mostly dark-colored structural carbon.

The quality of the boards was evaluated using an exact measure to determine their mechanical toughness, understood here as the integrated product of applied stress and strain per unit of cross section of the investigated board, i.e., the work of flexural test *W_f_.* The latter parameter can be determined as the work made over the deflection curve during the bending test [12]. In this study, the authors began testing with an initial force *F_0_* of 2 N and continued to break the fiber reinforcements so that the final decrease was registered at 40% of the maximum load—*F*_0.4 *MAX*_. Under these conditions, the flexural test *W_f_* can be calculated by applying the formula:(1)$$W_{f}\frac{1}{\;S\;}\int_{F_{0}}^{F_{0.4\;MAX}}F\;da$$
where *S* = specimen cross section; *a* = specimen deflection under the loading pin.

The loss of FCB elasticity can also be determined by applying the ultrasound testing (UT) method. In large objects with small thicknesses, like the flat boards, the following dependence combines UT longitudinal wave velocity *c_L_* and the modulus of elasticity *E*:(2)$$c_{L}=\sqrt{\frac{E}{\rho \left(1-\nu^{2}\right)}}$$
where *ρ* = bulk material density; *ν* = Poisson ratio.

## 2. Materials and Methods

In this study, five sets of specimens, made of five different types of FCB, were prepared for examination. These were labeled A, B, C, D and E. All of the specimens were fabricated by applying the Hatschek, or flow-on, forming method. Different matrix fillers and concentrations of fibers were determined for each set and resulted in different flexural strengths, which was measured by applying the standard EN 12467 three-point bending test [18]. The specimens of all FCB types, listed above, were stored in a dry, electric oven for 3 h at a temperature of 230 °C. The parameters of the heating procedure were chosen experimentally, after performing some preliminary tests to evoke considerable changes in the microstructure of the investigated materials. That treatment resulted in the decomposition (i.e., numerous broken chemical C–O bonds) of fiber reinforcements due to the process of carbonization. The carbonization process mostly concerns the linear chains of dehydrated glucose molecules, which are responsible for building up the cellulose fiber system.

The specimens, which underwent the high temperature treatment, were labeled A_T_, B_T_, C_T_, D_T_ and E_T_. Small pieces of each FCB type were prepared for microscopic observation, both before and after the elevated temperature treatment, and are presented in Figure 2. The mechanical properties of the specimens are shown in Table 1.

For detailed insight into each specimen’s microstructure, the authors applied an X-ray microtomography (micro-CT) technique, which is described in more detail in [13]. A Nanotom 30, made by General Electric (Baker Hughes GE, Houston, TX, USA) was used in the investigation. The system included a micro-focal source of X-ray radiation, a movable table on which to place a specimen, and a flat FCB panel with a radiation detector, having a resolution of 2000 × 2000 pixels. The microstructure of each FCB sample was observed on the cross sections (tomograms) of the investigated specimens, using a grey scale convention that was directly related to the amount of the local radiation absorption of the materials. The grey scale covers a wide range of grey levels and is ordered from pure white, related to maximum absorption, to pure black, related to minimum absorption, respectively. Un-hydrated cement particles and aggregate grains are objects in the cement matrix that demonstrate the highest degree of absorption ability. The hydration products that make up the major components of the cement matrix present a slightly lower absorption ability. Next in line are the hydrated calcinates, which demonstrate an even lower absorption ability and, at the end of the scale, are the organic fibers (if present) and the regions of high porosity. To obtain the optimal X-ray penetration and absorption of the investigated specimens, the following parameters of the scanning procedure were set: X-ray lamp voltage—115 kV, lamp current—95 microamperes, and shot exposition time—750 ms. Scanning, performed by the authors in this study, resulted in a large set of tomograms (specimen cross sections), performed for every 5 µm of the specimen height.

The authors prepared a dedicated instrumentation set to determine the UT longitudinal wave velocity c*_L_* in the boards made of fibrous materials. The Wave velocity c*_L_* was determined using the UT material tester, which was capable of measuring the time of flight *T* of the elastic wave front, across a board of known thickness *d,* with the application of the formula *c_L_* = *d/T*. The investigation was done using the ultrasonic material tester, UTC110, produced by Eurosonic (Vitrolles, France) [19]. A report in the related literature indicates that low-frequency ultrasound (50–200 kHz) was routinely used to characterize defects (a few centimeters in size) in the concrete structures on site. However, ultrasound at low-frequency ranges cannot be used to test fiber cement boards. Some experiments [11] have revealed that the sensitivity of the ultrasound parameters required to determine the structural properties of FCB is achieved when the ultrasound wavelength becomes comparable to the dimensions of the local delaminations and the lengths of the fiber inclusions. This wavelength *λ* remains in the following relation to the frequency *f* of the emitting source and the propagation velocity of the traveling ultrasonic longitudinal waveform:(3)$$\lambda =\frac{c_{L}}{f}$$

Thus, taking into account the propagation velocity of 1000–2000 m/s registered in the FCB, the authors recommend the application of a frequency of 1 MHz to achieve the propagation of wavelengths in the range of 1–2 mm. The instrumentation included an Olympus Videoscan [20] transmitting and receiving transducer, which emitted an ultrasonic beam measuring 19 mm in diameter, at a frequency of 1 MHz. The parameters were designed for coupling with low-density (i.e., 1000–2000 kg/m^3^) materials and, thus, exhibited a low-acoustic impedance of 10 MegaRayl. The contact between the rough surface of the FCB and the face of the transducer was achieved by using a 0.6 mm thick layer of polymer jelly interfacing foil (PM-4-12) produced by Olympus (Waltham, MA, USA) [20]. The custom-designed holder, with articulated joints and a compression spring, was prepared to ensure the correct coupling of the ultrasonic transducers to both surfaces of the investigated boards. A detailed view of the holder is presented in Figure 3.

## 3. Results

### 3.1. Optical Light Microscopy 

The morphology and surface views of all of the investigated compositions were analysed by visual light microscopy (AM4113ZTL 1.3 Megapixel Dino-Lite Digital Microscope with integral LED lighting, AnMo Electronics Corp. (Hsinchu, Taiwan). The magnified surface views of the investigated specimens are presented in Figure 4. Highly-diverse fiber distributions, due to the different fiber compositions, are noticeable in the micrographs. 

### 3.2. Scanning Electron Microscopy

A high-resolution environmental scanning electron microscope, Quanta 250 FEG, FEI (Hillsboro, OR, USA), was used in the investigations, along with an energy dispersive X-ray spectroscopy (EDS) analyzer. Figure 5 shows the exemplary SEM images for the tested fiber cement boards. Samples that were not exposed to a high temperature are shown on the left, while those that were exposed to a temperature of 230 °C are shown on the right. Exemplary elemental composition results for board C, which were obtained using the EDS analyzer, are shown in Figure 6.

An analysis of the images obtained from the scanning electron microscope and the EDS analyzer shows that the fiber cement boards, in the A–E series, have a compact macrostructure. Microscopic examinations revealed a fine-pore structure, with pores of up to 50 µm in size. Cavities and grooves, up to 500 µm wide, were visible in the fracture areas where the fibers had been pulled out. Cellulose fibers and, in some boards (B and D), PVA fibers, are clearly visible in the images. Various forms of hydrated calcium silicates of the C-S-H type occur. Both an “amorphous” phase and a phase built of strongly-adhering particles predominate. An analysis of the fiber composition showed that fiber elements and some cement elements were present. An analysis of the chemical composition of the matrix showed elements that are typical of cement. The surface of the fibers was covered with a thin layer of cement paste and hydration products. The fact that there are very few areas with a space between the fibers and the cement paste, indicates that the fiber-cement bond is strong.

A macroscopic analysis of the fiber cement boards in the **A_T_**–**E_T_** series, which were exposed to a temperature of 230 °C for 3 h, shows a clear change in the color of the samples. Examinations of the **A_T_**–**E_T_** fiber cement boards yielded consistent results. Most of the fibers in the boards were found to be burnt-out, or melted into the matrix, leaving cavities and grooves which were visible in all of the tested boards. The structure of the few remaining fibers was strongly degraded. Examination of the cement particles in further fractures revealed that burning-out also degrades their structure. The structure of the matrix was found to be more granular, showing many delaminations. Numerous caverns and grooves left by the pulled-out fibers, as well as the pulled-out cement particles, were observed.

### 3.3. X-ray Microtomography

Examples of virtually-cut, three-dimensional projections of 4 × 4 × 4 mm^3^ cubes, from specimens D, D_T_ and B_T,_ are presented in Figure 7. Specimen D, i.e., before the elevated temperature treatment, shows the regular microstructure without faults such as delaminations. The results of the elevated temperature impact is visible within the volume of specimen D_T,_ in the form of significant delaminations. The size and number of these delaminations are even more visible in the image of specimen B_T_, which was made of a material with lower mechanical performance.

Another way to present information about the specimens, derived from the X-ray scanning procedure, is to determine the brightness distribution of all of the examined voxels (i.e., volumetric pixels). The magnitudes of brightness of the different microstructural elements were included in the following ranges: the area of voids and fibers: 0–50 arbitrary units, a.u., fillers: 50–140 a.u., and dense phases; i.e., un-hydrated cement and fine aggregate grains: 140–170 a.u. The greyscale brightness distribution (GBD) of all voxels belonging to the three specimens presented in Figure 7, are shown in Figure 8. It is worth noting that the occurrence of the delamination processes caused a shift of the affected GBD curves to the left, i.e., in the direction of a region of voids.

### 3.4. Assessment of Ultrasound Wave Velocity c_L_ Loss and Presentation of the Results of Mechanical Tests

The results of the UT longitudinal wave velocity c_L_ tests are presented in Figure 9. For each 30 × 30 cm^2^ FCB board (A, B, C, D, E, A_T_, B_T_, C_T_, D_T_ and E_T_), two series, of ten measurements each, were performed in order to determine the dispersion of the results. The measurements were done, randomly, at different locations, over the entire surface of the boards. The standard deviation of a single series of measurements was included in the range of 2–3% of the average value of the readings. The UT longitudinal wave velocity c_L_ loss was observed in all cases of thermal treatment; however, that loss varied from 2% to 20% depending on the FCB composition. It is worth mentioning that the time to perform ten UT measurements took approximately 5 min, suggesting that this may be a good method for in-situ application.

To determine the changes in the mechanical toughness of the specimens, the authors performed bending tests using three 30 × 30 cm^2^ samples of each kind of board, following the requirements of the ISO 8336 [21]. The average results of the mechanical tests performed on the three samples are presented in Table 2.

Figure 10 and Table 2 present the results of the mechanical tests. The blue curve shows the behavior of the specimens in an as-delivered state, and the red curve shows the behavior of the specimens with the pyrolyzed reinforcing fibers. Based on an analysis of the curves, the board specimens in an as-delivered state are capable of withstanding a load close to F_MAX_ for the time required to destroy the fiber reinforcement system. The energy required for that damage can be estimated as W_f_, by applying Formula (1). The specimens that underwent the cellulose fiber carbonizing process demonstrated the brittle characteristics of the rupture, i.e., the break of the material cross section appeared immediately after reaching the critical F_MAX_ level. Work of flexural test, calculated for the carbonized specimens, equaled approximately 2–5% of the value estimated for the as-delivered state of the FCB specimens. It is also worth mentioning that, in the composition of boards with a low cement content, the brittle matrix broke under a relatively low loading force, while its internal fiber reinforcement system could withstand more stages of the damage process.

## 4. Conclusions

The authors investigated five different compositions of fiber cement boards. Two of these compositions, A and B, contained a low amount of cement, which resulted in low flexural strength (14–21 MPa). The other three compositions contained more cement and their flexural strength was determined at the higher range of 23–36 MPa. The fibers applied in compositions A and D, having the best quality and proper length (approximately 3 mm), resulted in the highest value of the work of flexural test *W_f_*, before carbonization. The carbonization process, designed in the laboratory to simulate the long exposure of FCBs to environmental hazards, significantly influenced the mechanical properties of all of the investigated compositions. The micrograph images of the carbonized specimens show the transition of the fibers from their original color into brown. The SEM examinations confirmed the marked changes in the structure that took place as a result of the exposure to a temperature of 230 °C for 3 h. In all of the tested fiber boards, most of the fibers were found to be burnt out, or melted into the matrix, leaving cavities and grooves. The structure of the few remaining fibers was highly degraded. The decrease of the *W_f_* parameter was considerable for all of the tested compositions, as a result of the embrittlement of the fiber reinforcements. The delaminations within the microstructure of the specimens, due to the thermal treatment, was clearly visible in the three-dimensional projections obtained by applying the micro-CT technique. The delaminations also caused a shift in the affected GBD curves to the left, i.e., into the region signalling the presence of loose phases.

In the opinion of the authors, the ultrasound method has proven its applicability for testing the quality of fiber cement boards. The dedicated UT transducers, with low acoustic impedance and polymer jelly interface, were capable of achieving the required propagation of UT waves in order to determine their velocity in the investigated materials. The UT wave velocity *c_L_* in compositions with low levels of flexural strength (A, B) was in the range of 1.1–1.6 km/s (1100–1600 m/s), whereas the wave velocity *c_L_* in compositions with higher flexural strength (B, C, D, E) was in the range of 1.7–2.22 km/s (1700–2220 m/s). The decrease of wave velocity *c_L_* after carbonization occurred in all tested compositions; however, its magnitude was diverse and was included in the range of 2–20%, in relative units. The lowest decrease of *c_L_* occurred in the board made with the best quality components, i.e., the board intended for external use. All of these characteristics lead the authors to recommend the UT method as a useful tool for the on-site assessment of the degradation processes occurring in fiber cement boards.

## Figures and Tables

**Figure 1 materials-12-00944-f001:**
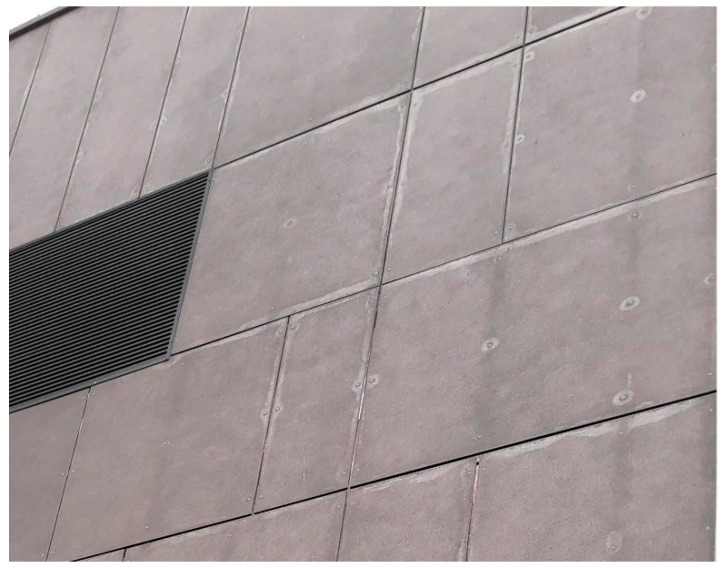
An example of a FCB façade showing considerable damage after ten years of exposure to climate hazards.

**Figure 2 materials-12-00944-f002:**
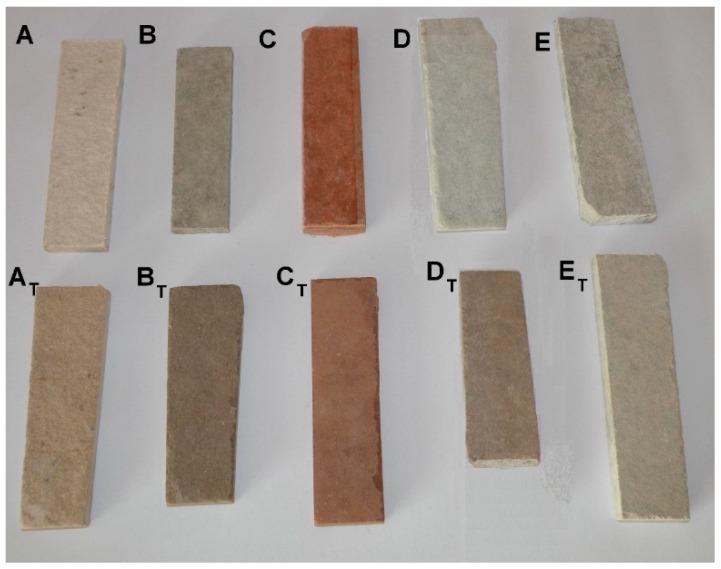
View of the small FCB (fiber cement board) pieces used for microscopic observation. Upper row: specimens of FCB types A, B, C, D and E before the elevated temperature treatment. Lower row: specimens of FCB types A, B, C, D and E after the elevated temperature treatment.

**Figure 3 materials-12-00944-f003:**
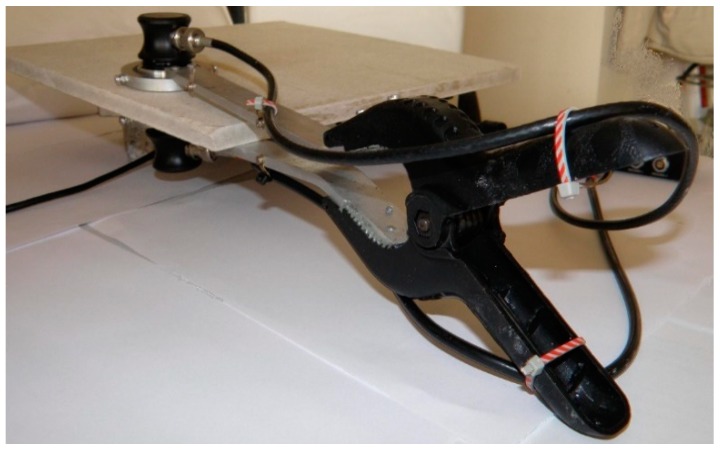
Detailed view of the custom-designed holder for correct coupling between the ultrasonic transducers and both sides of the rough surface board.

**Figure 4 materials-12-00944-f004:**
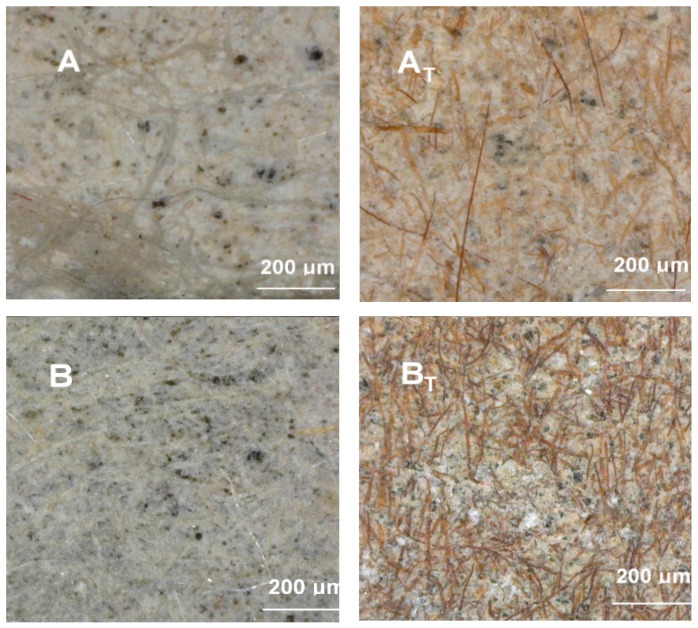
Micrographs of the surface views of the investigated specimens. **Left**: before the elevated temperature treatment; **Right**: after the elevated temperature treatment.

**Figure 5 materials-12-00944-f005:**
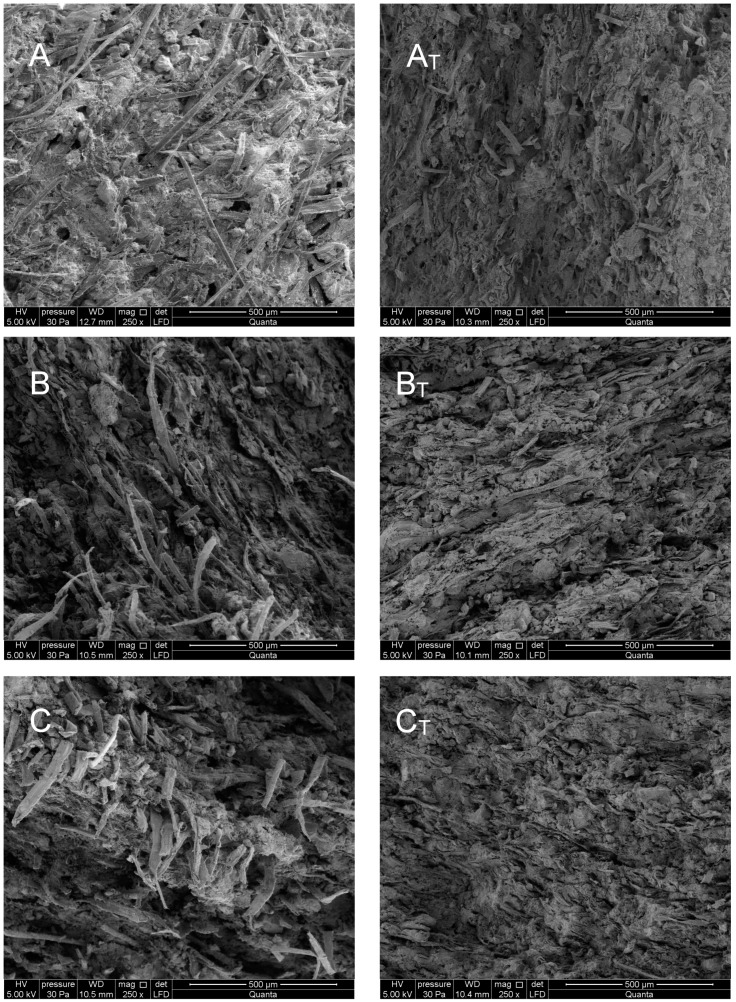
SEM images for boards (**A**–**E**) (left) and (**A_T_**–**E_T_**) (right).

**Figure 6 materials-12-00944-f006:**
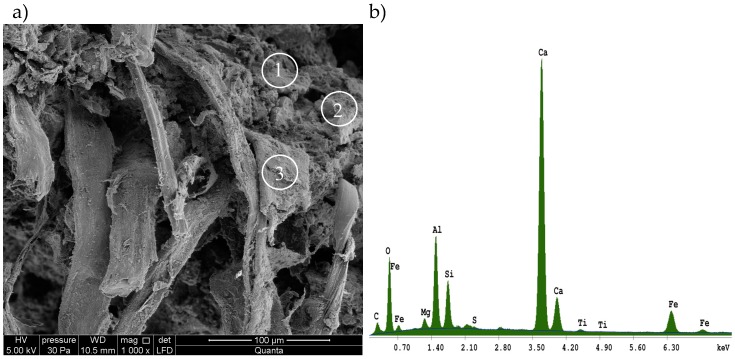
Results obtained using the energy dispersive X-ray spectroscopy (EDS) analyzer for board C: (**a**) areas of elemental composition analysis, (**b**) results of EDS in point 1, (**c**) results of EDS in point 2, (**d**) results of EDS in point 3.

**Figure 7 materials-12-00944-f007:**
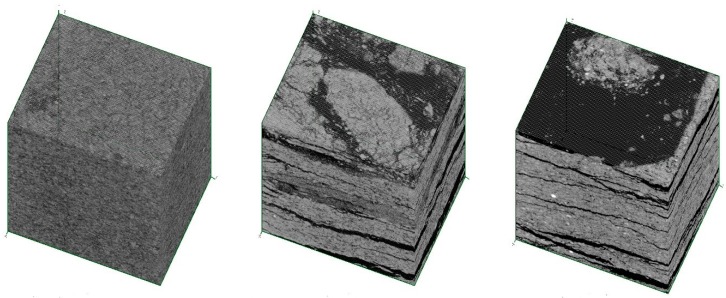
Examples of virtually-cut, three-dimensional projections of 4 × 4 × 4 mm^3^ cubes from specimens “D” (**left**), “D_T_” (**center**) and “B_T_” (**right**). Specimen “D” shows the regular microstructure without faults such as delaminations. The results of the elevated temperature impact is visible within the volume of specimen “D_T_” and “B_T_” in the form of significant delaminations.

**Figure 8 materials-12-00944-f008:**
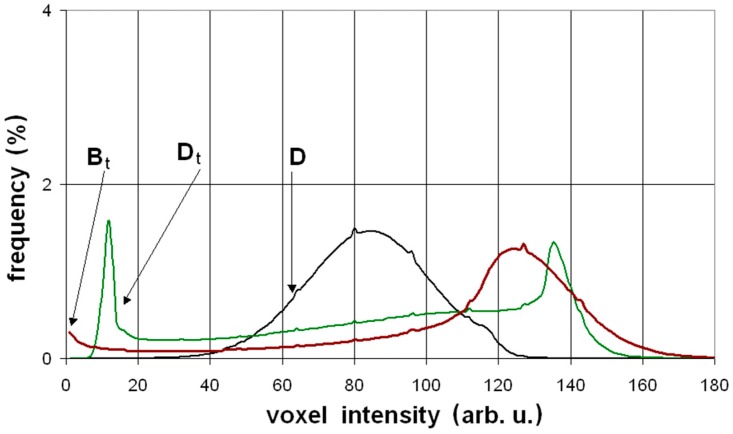
The greyscale brightness distribution (GBD) of all voxels belonging to the three examined specimens: “D”, “D_T_” and “B_T_”. The occurrence of the delamination processes caused a shift of the affected GBD curves to the left, i.e., in the direction of a region of voids.

**Figure 9 materials-12-00944-f009:**
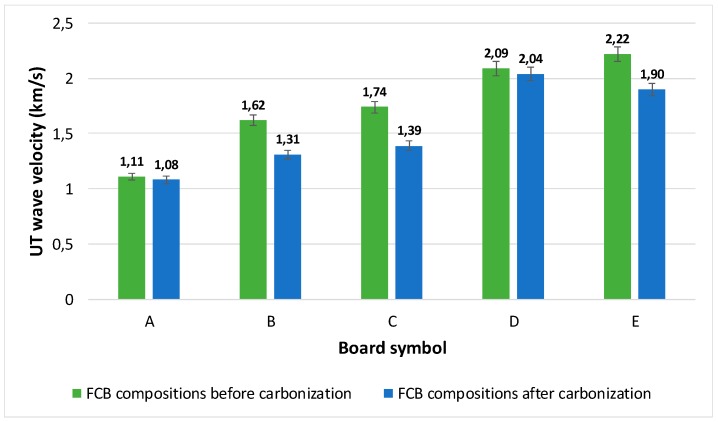
Results of the UT longitudinal wave velocity c_L_ tests, measured in all FCB compositions, before and after carbonization. The black marks depict standard deviations (2–3%), determined in populations of the measurements.

**Figure 10 materials-12-00944-f010:**
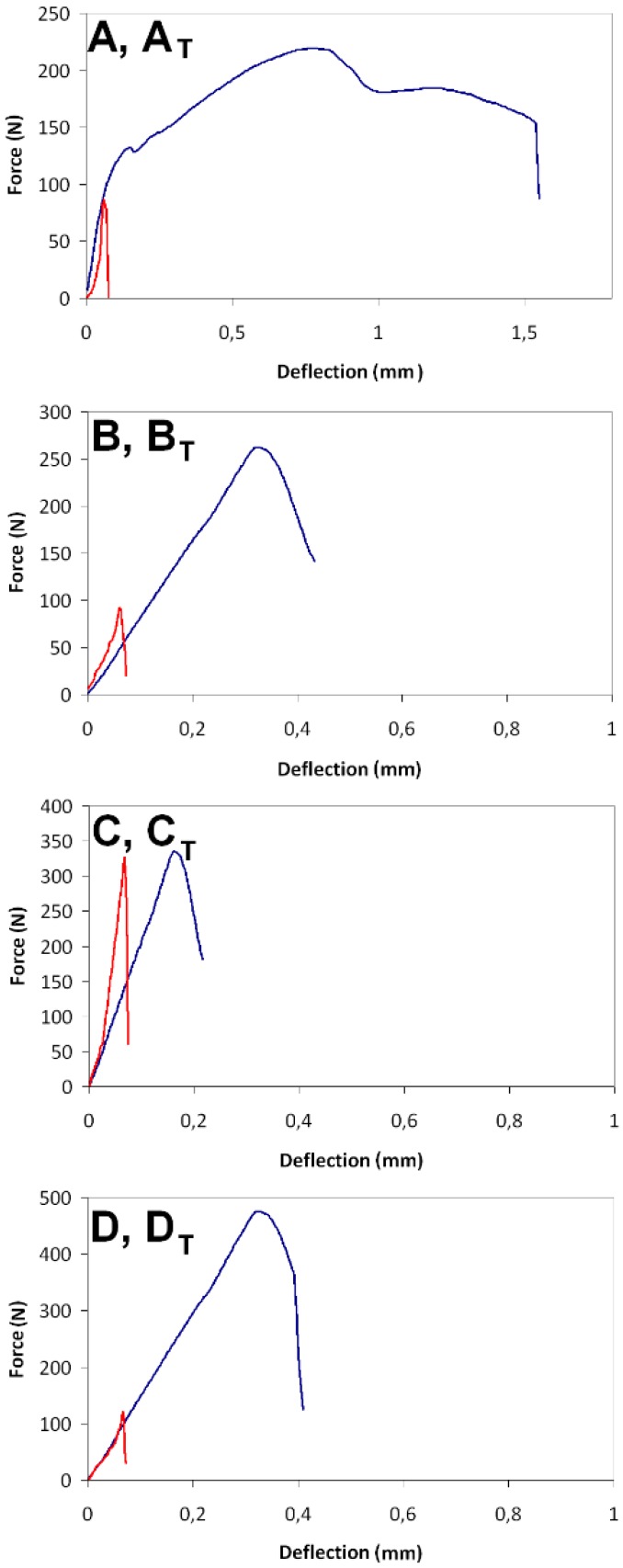
Typical load-deflection curves of the tested FCB compositions. The blue curves show the behavior of the specimen in an as-delivered state and the red curves show the behavior of the specimens with the carbonized reinforcing fibers.

**Table 1 materials-12-00944-t001:** Mechanical properties of the tested FCB compositions.

Board Symbol	Remarks	Board Thickness [mm]	Apparent Density [kg/m^3^]	Flexural Strength [MPa]
A	low cement content in the board matrix, approved for internal use	8.4	1000	14
B	low cement content in the board matrix, approved for internal use	7.4	1600	21
C	colored brown and approved for internal use	7.8	1700	23
D	approved for external use	8	1600	36
E	approved for internal use	9.3	1700	23

**Table 2 materials-12-00944-t002:** The results of the mechanical tests performed on the investigated FCB compositions.

Board Symbol	F_MAX_ in the State As-Delivered [N]	F_MAX_ after Thermal Treatment [N]	Work of Flexural Test *W_f_* in the State As-Delivered [J/m^2^]	Work of Flexural Test *W_f_* after Thermal Treatment [J/m^2^]
A	216	78	1330	29
B	250	82	681	32
C	475	115	848	190
D	330	384	1104	37
E	424	446	790	331

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
