# Peer review of "Investigation of Structural Degradation of Fiber Cement Boards Due to Thermal Impact"

_materials, 2019, doi:10.3390/ma12060944_

Round 1

Reviewer 1 Report

The submitted manuscript examines the degradation of microstructure and mechanical properties of fibre cement boards exposed to environmental hazards/thermal impact. In my opinion, the presented results are interesting and the paper is well structured. Nonetheless, I would like to bring some minor issues to the attention of the authors, before the paper be accepted for publication. As such, I suggest revision of the original manuscript, according to the following comments:

1) Line 31: please repeat the acronym here.

2) Lines 38-39: please rephrase so as to avoid using “one” in a technical document. The same applies with line 72 “…can be found in [17]”.

3) Line 92-94: please add this standard to the reference list and also explain the reason why the authors have considered 3 hours at 230° C. Explanations could help the reader.

4) Lines 115-116: state why – please expand.

5) Line 146: please amend typo (ot).

6) Lines 203-205: please amend/rephrase this sentence.

7) Line 228: I would suggest replacing “what” with “something which”.

8) Section 3.3: please compute and compare the energies associated with the plots of Figure 10. Discussion should come along with these numbers.

Author Response

We are deeply grateful to the Reviewer for the effort put in the review of our paper.

We agree with most of the Reviewer’s comments and we have taken them into account in the paper’s revised version.

1) Line 31: please repeat the acronym here. - CORRECTED

2) Lines 38-39: please rephrase so as to avoid using “one” in a technical document. The same applies with line 72 “…can be found in [17]”. - CORRECTED

3) Line 92-94: please add this standard to the reference list and also explain the reason why the authors have considered 3 hours at 230° C. Explanations could help the reader. - CORRECTED

4) Lines 115-116: state why – please expand. - CORRECTED

5) Line 146: please amend typo (ot).  - CORRECTED

6) Lines 203-205: please amend/rephrase this sentence. - CORRECTED

7) Line 228: I would suggest replacing “what” with “something which”. - CORRECTED

The authors are convinced that many of the Reviewer’s suggestions will be helpful in further research and analyses which will form the basis for the next paper.

Once again we would like to thank the Reviewer most warmly for the perceptive and detailed comments, which greatly enhance the understanding of the paper and its value.

Reviewer 2 Report

Positive properties:

The study is focused on investigation of stability and durability problem of fiber cement boards (FCB). Used methods of X-ray micro tomography 104 (micro-CT) and ultrasonic investigation are innovative for FCF testing.

Basic remarks:

English style and grammar should be checked and improved, some phrases are not understandable, for example “ices clods hits” (page 1, raw 39), etc.

Explain please what means carbonization in temperature of 230 oC (raw 94: “That treatment resulted in decomposition of fiber reinforcement due to process of carbonization”.

In the abstract and introduction it was stated:  “The aim of presented research was to investigate the degradation of microstructure and mechanical properties of fiber cement boards (FCB) exposed to environmental hazards: visual & ultraviolet light radiation or thermal stresses”.  In conclusion: “The laboratory made carbonization process intended to simulate the long exploitation under environmental hazards...”.  In the study the samples were subjected only to thermal treatment at 230oC. In this case it can’t be climbed that high temperature impact simulates long-term exploitation conditions of FCB.

Therefore, the aim of the study should be formulated as investigation of structural degradation of fiber cement boards due to thermal impact.

Cement and wood compositions usually are sensitive to moisture impact, but the effect of external moisture is not mentioned among adverse factors. Furthermore, moisture is very important factor when explaining processes of degradation of wood and cement based composites. Analysis of result should be focused also on explanation of processes and corresponding mechanisms during thermal treatment (dehydration, shrinkage, vapour internal pressure, delamination etc).

Author Response

We are deeply grateful to the Reviewer for the effort put in the review of our paper.

We agree with most of the Reviewer’s comments and we have taken them into account in the paper’s revised version.

English style and grammar should be checked and improved, some phrases are not understandable, for example “ices clods hits” (page 1, raw 39), etc.

The linguistic errors have been corrected. The paper has been checked by a sworn translator of the English language.

Explain please what means carbonization in temperature of 230 oC (raw 94: “That treatment resulted in decomposition of fiber reinforcement due to process of carbonization”.

The term carbonization in temperature of 230 oC was explained.

Cement and wood compositions usually are sensitive to moisture impact, but the effect of external moisture is not mentioned among adverse factors.

Thank you of this advice. This is the topic of our next article.

The authors are convinced that many of the Reviewer’s suggestions will be helpful in further research and analyses which will form the basis for the next paper.

Once again we would like to thank the Reviewer most warmly for the perceptive and detailed comments, which greatly enhance the understanding of the paper and its value.